# An Investigation of the Anisotropic Mechanical Properties of Additive-Manufactured 316L SS with SLM

**DOI:** 10.3390/ma17092017

**Published:** 2024-04-26

**Authors:** Haibo Wang, Peng Jiang, Guangyong Yang, Yu Yan

**Affiliations:** School of Mechanical and Materials Engineering, North China University of Technology, Beijing 100144, China; wanghaibo@ncut.edu.cn (H.W.); jiangpeng.sd@foxmail.com (P.J.); 13691340818@163.com (G.Y.)

**Keywords:** SLM, mechanical properties, anisotropic, EBSD

## Abstract

Selective laser melting (SLM) forms specimens that often exhibit anisotropic mechanical properties. Most existing research only explains that the mechanical properties of specimens perpendicular to the build direction are superior to those parallel to the build direction. In this paper, the mechanical properties of SLM 316L SS specimens with different surfaces and different directions are compared. Finally, it was found that the mechanical properties of specimens on Face 3 are stronger than those on Face 1 and Face 2, while the mechanical properties of specimens on Face 1 and Face 2 are similar. For specimens in different directions on the same surface, the mechanical properties of Face 1 and Face 2 exhibit clear anisotropy, while the mechanical properties of Face 3 tend to be isotropic. In this paper, the EBSD technique was used to analyze the specimens. It was found that the anisotropy of the mechanical properties of Face 1 and Face 2 are attributed to the presence of texture and columnar crystals in the sample. This paper can provide accurate and reliable material performance data for the practical application of SLM 316L SS, thereby guiding the optimization of engineering design and manufacturing processes.

## 1. Introduction

316L SS is an austenitic stainless steel with a wide range of applications. The composition (in wt.%) of grade 316L SS is as follows: Cr (16–18), Ni (10–14), Mo (2–3), C (≤0.03), Mn (≤2), Si (≤0.75), P (≤0.05), S (≤0.03), N (≤0.1), and Fe (Balance); the ‘L’ denotes low carbon concentration [1]. 316L SS has been recognized as an important alloy for a wide range of industrial applications, from household goods to aerospace and nuclear industries, due to its superior properties such as high ductility, weldability, medium yield strength, high corrosion resistance, and relatively low cost [2]. SLM, an emerging additive manufacturing (AM) technology, is receiving increasing attention due to its superior ability to produce high-performance parts with complex geometries [3,4]. Nowadays, SLM has been widely used in the manufacturing of various 316L SS parts. Some specialized applications, such as directionally solidified turbine blades, require anisotropic properties along specific planes. However, anisotropic mechanical properties might be undesirable for structural applications [5]. In order to provide more accurate and reliable material performance data for practical applications and to guide the optimization of engineering design and manufacturing processes, it is necessary to conduct a more comprehensive study on the anisotropy of the mechanical behavior of SLM 316L SS.

In recent years, numerous studies have demonstrated that in laser additive manufacturing (LAM), the strength of samples perpendicular to the building direction is higher than that of samples parallel to the building direction [6,7,8,9,10,11,12,13]. 

In order to obtain a more comprehensive understanding of the anisotropic performance of mechanical properties, researchers have studied the mechanical properties of AM 316L SS with different directions and different surfaces. V. Ajay et al. [5] studied wire arc additively manufactured 316L SS and found that the columnar grain growth along the build direction results in a {001} <100> crystal texture in the deposit. The crystal texture negatively affects the elastic properties, leading to the lowest yield strength and elastic modulus being achieved along the build direction. The tensile test performance along the diagonal direction is significantly superior to that along the build direction. Lan Kang et al. [14] studied the mechanical properties and microstructure of LAM 316L SS and found that the mechanical properties of LAM 316L SS on the plane parallel to the building direction exhibit noticeable anisotropy. The microstructure analyses revealed a distinct layered structure perpendicular to the building direction and some columnar crystals perpendicular to the layer plane crossing the boundary, which might explain the reasons causing the high anisotropy of the macroscopic mechanical properties of LAM steels. Som Dixit et al. [15] studied LPBF 316L SS and found that the strength of the sample in the horizontal direction was higher than that of the sample in the vertical direction, while the ductility in the Z direction sample was the highest. The XY45° sample exhibited higher YS and UTS than the X and Y samples at all levels. Crystal orientation and dislocation density are the primary factors contributing to the anisotropy of mechanical properties.

Similar results have been found in the study of other materials. Beibei He et al. found that the anisotropy of the mechanical properties of the XZ plane in SLM Ti6Al4V specimens is more obvious than that of the XY plane [16]. Yupeng Ren et al. studied the mechanical properties of cold-sprayed AM 7075 Al coating and found that the mechanical anisotropy primarily existed between the deposition direction (Z) and the XY plane. When the tensile direction changes from the X/Y direction to the Z direction, the ultimate tensile strength gradually decreases [17]. A. Charmi et al. found that the yield strength of the specimen increased as the angle between the specimen sampling direction and the substrate decreased [18]. L. Palmeira Belotti et al. studied the anisotropy of stainless steel parts fabricated by thick-walled arc additive manufacturing, and found that under uniaxial tensile loading conditions, the material exhibited an anisotropic mechanical response in the TDBD plane along the transverse, 45° from the transverse (diagonal), and BD directions [19]. L. Hitzler et al. found that the variation in mechanical characteristics resulting from altering the inclination between the loading and the layers differed, and was demonstrated to be highly dependent on the material [20].

In existing research, it has been found that there are different variations of anisotropy between the plane perpendicular to the construction direction and the plane parallel to the construction direction in AM technology. Researchers attribute the anisotropy of mechanical properties to manufacturing defects [21], textures [15,18,22,23,24,25], dislocations [15], and grain size and shape (columnar crystals) [26,27,28], among other factors. 

When manufacturing parts, we can determine the optimal direction based on the material’s anisotropy in order to achieve the necessary mechanical properties. However, existing research lacks a comprehensive description and micro-interpretation of the anisotropy of mechanical properties of SLM 316L SS samples in different planes and directions. Therefore, it is necessary to study the anisotropy of SLM 316L SS. This paper analyzes the mechanical properties of SLM 316L SS from different planes and directions, and establishes the anisotropic laws of these properties. The causes of the anisotropic mechanical properties were investigated using the EBSD technique, focusing on grain size and shape, dislocation, and texture. Finally, through the comparison of the mechanical properties of the fabricated specimens, the most favorable combination of printing parameters is identified in this paper, offering a reference to other researchers in the selection of optimal process parameters for SLM 316L SS.

## 2. Materials and Methods

In this study, the 316L SS powder used was provided by Beijing e-Plus 3D Tech. Co., Ltd., Beijing, China. In the SLM process, the physical properties of the powder, such as particle size distribution, fluidity, and bulk density, significantly influence the structure, shape, and mechanical properties of the specimen. First, the chemical composition of the powder will directly influence the density and chemical stability of the formed parts, as well as the chemical properties of the formed parts. Second, the particle size distribution of the powder will influence the surface smoothness and strength of the formed part. Bulk density will influence the porosity and compactness during the sintering process. The flowability will directly influence the accuracy of forming and the surface quality during the forming process. The oxygen content will influence the extent of oxidation and the mechanical properties of the formed parts. The sphericity of the powder also influences the sintering process of the formed part, as well as the final density and surface quality. The chemical composition of the 316L SS powder used in this experiment is shown in Table 1, and the physical properties of the powder are shown in Table 2.

The LAM equipment used in this paper is the EP-M150 small-size mainstream metal AM equipment produced by Beijing e-Plus 3D Tech. Co., Ltd., Beijing, China. as shown in Figure 1. The equipment includes a printing system, a control system, and a filtering system. The printing system utilizes a maximum laser power of 500 W, with a minimum spot diameter of the laser beam at 40 μm, and a maximum laser scanning speed of 8000 mm/s. The size of the forming chamber of the printing system is Φ153 × 100 mm^3^. In order to prevent oxidation of the metal powder during the forming process, a sealed cavity is utilized, and argon gas is introduced as a protective gas, with an oxygen content of less than 0.02%.

In the LAM process, the quality of formed products is mainly influenced by factors such as laser power, scanning spacing, scanning speed, and scanning strategy. These factors will significantly impact the surface and internal microstructure of the formed parts, thereby significantly affecting the quality of the formed parts. This paper investigates the impact of laser power, scanning speed, and scanning spacing on the mechanical properties of the sample, without delving deeply into the scanning strategy and powder thickness.

In Liang Hao’s [29] study, it was observed that 316L samples, formed at various laser power and scanning speed settings, exhibited high forming rates and hardness when the laser power ranged from 200 to 300 W, coupled with scanning speeds of 600 mm/s, 900 mm/s, and 1100 mm/s. In line with the optimal parameters for SLM 316L SS identified by Baris Sener, the scanning spacing was set to 0.1 mm [30]. Therefore, the laser power of this experiment was set to 200–300 W, the scanning speed was set to 850–1000 mm/s, and the scanning spacing was set to 0.08–0.14 mm. The scanning strategy employed in the experiment involved deflecting the adjacent layer by 67°, with a power layer thickness of 0.03 mm. The specific parameter settings for this experiment are shown in Table 3.

The printed sample and pick-up position of the SLM are shown in Figure 2. In this paper, we refer to the XY plane as Face 3, which is perpendicular to the building direction. Additionally, we refer to the XZ plane as Face 1 and the YZ plane as Face 2; both are parallel to the building direction and perpendicular to each other. Specimens were taken from each surface in three directions: 0°, 45°, and 90°, with two specimens taken in each direction. The mechanical properties of the two samples taken from the same position are similar, and the stress-strain curves derived from the experimental data also exhibit similarity. Therefore, the stress-strain curves in the Results and Discussions section are derived from the data of the sample with superior mechanical properties.

The numbering rule for uniaxial tensile specimens follows the format of ‘process parameters–plane of the pick-up–direction of the pick-up’. For example, in ‘1-2-DH’, ‘1’ represents the parameter selected from Table 3 for the printing process, ‘2’ indicates that the specimen is taken from Face 2 of Figure 2, ‘D’ denotes that the specimen is a uniaxial tensile specimen, and ‘H’ indicates the direction of the specimen. There are three forms of ‘H, T, and V’ representing the directions of the sample. ‘H’ corresponds to 0°, ‘T’ corresponds to 45°, and ‘V’ corresponds to 90°, as shown in Figure 2. The specimen’s shape design and the static tensile test method for metal materials refer to GB/T 228-2010. The specific size of the uniaxial tensile specimen is shown in Figure 3, and the specimen is obtained by wire cutting.

In this paper, the mechanical properties of materials are obtained through uniaxial tensile testing. The uniaxial tensile equipment used in this study is the Instron 5982 universal testing machine manufactured by Instron Company, Chicago, IL, USA, as shown in Figure 4. The maximum load can reach 100 KN. The strain rate used in the uniaxial tensile test is 0.035 mm/s. 

Through uniaxial tensile testing the load-displacement data of SLM 316L SS can be obtained, and the test used DIC equipment to measure the strain of 316L SS. The displacement and load data obtained from the tensile equipment were imported into the DIC equipment calculation software Bluehill Universal 4.08 to calculate engineering stress and engineering strain data. Subsequently, the data can be processed by the following formula to obtain the corresponding true stress-strain curves of ten groups of tensile specimens with different process parameters. According to the formula, the true stress-strain curve can be obtained. The material’s elasticity modulus, yield strength, tensile strength, elongation, and other mechanical properties are further obtained according to the true stress-strain curve.

The equations for calculating the true stress and true strain are as follows: (1)$$\sigma_{0}=\frac{F}{A_{0}},$$
(2)$$\varepsilon_{0}=\frac{\Delta l}{l_{0}},$$
(3)$$\sigma_{t}=\sigma_{0}(1+\varepsilon_{0}),$$
(4)$$\varepsilon_{t}=\ln(1+\varepsilon_{0})$$
where *F* is the value of the tensile force during the tensile test; *A*_0_ is the original cross-sectional area of the tensile specimen at a distance from the specimen; Δ*l* is the amount of change in the length direction of the scalar segment of a tensile specimen during a tensile test; *l*_0_ is the original length of the scale section of the tensile specimen; *σ*_0_ and *ε*_0_ are engineering stress and engineering strain, respectively; and *σ*_*t*_ and *ε*_*t*_ are the true stress and true strain, respectively.

In this study, electron backscatter diffraction (EBSD) was used to observe the samples. The EBSD device is from Oxford, UK. The working parameters of the EBSD test were as follows: the test voltage was 20 kv, the step size was 2 um, and the sample inclination angle was 70°. The sample preparation method was mechanical polishing.

## 3. Results and Discussions

The mechanical properties of 316L stainless steel under different laser powers, scanning speeds, and scanning spacings were analyzed, in order to determine the anisotropic laws of its mechanical properties and explain the underlying causes of its mechanical behavior from a microscopic perspective. Comparing the mechanical properties of the specimens with different printing parameters, the printing parameters used in the specimens with better mechanical properties in this test were selected.

### 3.1. Anisotropic Laws of Mechanical Properties

Figure 5, Figure 6, Figure 7, Figure 8, Figure 9, Figure 10, Figure 11, Figure 12, Figure 13 and Figure 14 show the true stress-strain curves and mechanical properties corresponding to each process parameter. In each figure, (a), (b), and (c) show the true stress-strain curves in different directions of Face 1, Face 2, and Face 3 under the corresponding printing parameter, respectively, while (d) shows the mechanical properties of specimens under the corresponding printing parameter. The specific data of mechanical properties can also be obtained from Table A1.

From Figure 5, Figure 6, Figure 7, Figure 8, Figure 9, Figure 10, Figure 11, Figure 12, Figure 13 and Figure 14, it can be found that the true strain of most uniaxial tensile specimens exceeds 0.3. The uniaxial tension specimens with true strains below 0.2 were the T direction on Face 1 of parameter 2, the T and V directions on Face 2; the T direction on Face 3 of parameter 4; the V direction on Face 1 of parameter 5; all directions on Face 1 of parameter 6, the T and H directions on Face 3; the V direction on Face 1 parameter 8; the H and V directions on Face 2; and the V direction on Face 1 of parameter 9. Among them, the specimens with a true strain below 0.2 are mainly concentrated in the T and V directions on Faces 1 and 2. During the initial stages of printing of some specimens, black dots appeared during the powder spreading process, so we conjecture that the smaller true strain is the result of printing defects.

If the true stress-strain curves of the printed specimens in different directions on each surface are similar under different printing parameters, it shows that the mechanical properties of the surface are isotropic. If the true stress-strain curves are significantly different, it shows obvious anisotropy. Based on this judgment, from Figure 5, Figure 6, Figure 7, Figure 8, Figure 9, Figure 10, Figure 11, Figure 12, Figure 13 and Figure 14, we can observe that Face 1 and 2 of Parameters 1, 2, 3, 5, 7, 8, 9, and 10 show obvious anisotropy, whereas the anisotropy of Face 3 is less pronounced. The results for Parameters 4 and 6 are significantly different from the rest of the parameters. All faces of Parameter 4 exhibit anisotropy. Face 2 of parameter 6 exhibits isotropy, while Faces 1 and 3 exhibit anisotropy.

From this, we can observe that the anisotropy of SLM 316L SS is primarily evident within Faces 1 and 2 (parallel to the stacking direction), whereas Face 3 (perpendicular to the stacking direction) exhibits good isotropy.

To better visualize the anisotropic laws of the mechanical properties, we compare the mechanical properties in the form of differences. The values of the mechanical properties are shown in Table A1. Here, given a criterion, when the difference in yield strength, tensile strength, and elasticity modulus between two specimens is less than 25 MPa, 100 MPa, and 25 GPa, respectively, we consider that their mechanical properties are close. In Figure 15 and Figure 16, the value of our judgment criterion is represented by ‘b’, and the actual value of the difference in mechanical properties is represented by ‘a’. From this criterion, we can see that a < (−b) (shown in red) indicates that the difference between the two mechanical properties is large, and the former is weaker than the latter; −b < a < b (shown in green) indicates that the difference between the two mechanical properties is not large; and a > b (shown in blue) indicates that the difference between the two mechanical properties is large, and the former is stronger than the latter.

Figure 15a–d are drawn according to Table A2, Table A3, Table A4 and Table A5 in Appendix A, respectively. Through Figure 15a–d, the mechanical properties of specimens with different directions in the same face are analyzed. From Figure 15a, it can be observed that the mechanical properties in the H and T directions are similar, while the mechanical properties in the V direction are inferior. From Figure 15b, it can be observed that the mechanical properties in Face 1 are T, H, V in order from good to bad; from Figure 15c, it can be observed that the mechanical properties in Face 2 are ‘T, H, V’ in order from good to bad; and from Figure 15d, it can be observed that the mechanical properties in the H, T, and V directions are similar in the three faces. This is consistent with the result that we obtain from the true stress-strain diagram that Face 3 tends to be isotropic. Moreover, we can also see that the strength of the specimen in Faces 1 and 2 perpendicular to the build direction is higher than that of the specimen parallel to the build direction.

Figure 16a–d are drawn according to Table A6, Table A7, Table A8 and Table A9 in Appendix A, respectively. Through Figure 16a–d, the mechanical properties of specimens with the same direction in different faces are analyzed. From Figure 16a, we can observe that Faces 1 and 2 have similar strength, while Face 3 is stronger. From Figure 16b, the mechanical properties of the three faces are relatively similar in the H direction. From Figure 16c, it can be observed that Face 1 is slightly stronger than Faces 2 and 3 overall in the T direction, while Face 2 is relatively similar to Face 3. From Figure 16d, it can be observed that Face 3 is stronger than Faces 1 and 2 in the V direction, while Face 1 is relatively similar to Face 2.

### 3.2. Microstructural Study

This section will explain the reasons for the anisotropy of mechanical properties in this study from three aspects: grain size and shape, texture, and dislocation density. Because the mechanical properties of the specimens printed with 10 parameters are better (see Section 3.3), and the law of anisotropy of mechanical properties is consistent with the law found in Section 3.1, this paper selects three different planes of samples with Parameter 10 for microscopic observation.

According to the Hall–Petch relation, the strength of the material increases as the crystal size decreases [31,32]. Smaller crystal size means more grain boundaries, which are the obstacles to dislocation movement. With an increase in the number of grain boundaries, dislocation movement requires higher stress, resulting in increased strength [14]. From Figure 17a–f, it is found that the grain size of Face 3 is significantly smaller than that of Face 1 and Face 2. The average grain sizes of Face 1, Face 2, and Face 3 are 35.0 µm, 52.8 µm, and 27.9 µm, respectively. 

The dislocation structure is extremely important in adjusting the plastic deformation behavior of the sample [33]. The average orientation deviation can reflect the dislocation density [15]. The dislocation density increases with the increase in the average orientation deviation [15,34]. From Figure 17g–i, it can be seen that the dislocation density of Face 3 is significantly higher than that of Face 1 and Face 2. The average orientation deviations (θ_KAM_) of Face 1, Face 2, and Face 3 are 0.61°, 0.67°, and 1.00°, respectively. 

The mechanical properties of Face 3, as indicated by the grain size and dislocation density, are better than those of Face 1 and Face 2, which is consistent with the experimental results.

From Figure 18, it can be observed that Face 1 exhibits a weak {001} <110> plate texture and a weak <102>∥X silk texture. From Figure 19, it can be observed that Face 2 exhibits a strong {001} <100> texture, with a texture intensity of up to 11.88. From Figure 20, it can be observed that Face 3 exhibits a weak {001} <110> texture, with a texture intensity of up to 5.41. It is found that there are differences in mechanical properties along different directions of texture [15]. Meanwhile, influenced by the local maximum thermal gradient during the solidification process and the crystal orientation of the solidified material under the liquid melt [27,35,36], a columnar crystal formed at a 45° angle to the building direction on Face 1 and Face 2. The generation of columnar crystals results in varying grain boundary density and dislocation density, ultimately leading to anisotropy in the mechanical properties of the plane (Face 1 and Face 2) parallel to the construction direction. Therefore, we believe that the anisotropy of the mechanical properties of Face 1 and Face 2 are attributed to the presence of texture and columnar crystals in the sample.

### 3.3. Select Process Parameters

In this section, the yield strength, tensile strength, elastic modulus, and elongation of the samples produced using different process parameters will be compared. It will then identify the printing parameters corresponding to the samples with superior mechanical properties, providing a reference for other researchers to select the optimal process parameters.

The previous study indicates that, with the exception of Parameter 6, the specimens obtained on Face 3 exhibit minimal anisotropy and demonstrate excellent mechanical properties. Therefore, in this paper, the specimens form Face 2 of Parameter 6 and the specimens from Face 3 of the remaining parameters are selected for analysis.

#### 3.3.1. Optimal Combination of Laser Power

According to the process parameters specified in Table 3, Parameters 1, 2, 3, and 4 were selected to investigate the impact of different laser powers on the forming properties of SLM 316L SS. In these four parameter sets, the scanning spacing and scanning speed were fixed at 0.10 mm and 1000 mm/s, while the laser power for Parameters 1, 2, 3, and 4 were 200 W, 230 W, 270 W, and 300 W, respectively. The optimal laser power was determined by analyzing the tensile properties of specimens under different laser powers. The true stress-strain curves of the specimens on Face 3 of Parameters 1, 2, 3, and 4 can be found in Figure 5, Figure 6, Figure 7 and Figure 8. Figure 21 shows the elongation of the corresponding specimens.

From the perspective of the proximity of stress-strain curves, the stress-strain curves in the three directions of Parameter 3 are closer. This means that among the specimens printed with these four groups of parameters, the specimens printed with Parameter 3 tend to be more isotropic. In addition, according to the mechanical properties’ indexes, the average yield strength, tensile strength, and elastic modulus of specimens in different directions of Parameter 3 were 490.57 MPa (the largest in the four groups), 861.09 MPa (the largest in the four groups), and 119.46 MPa (the second largest in the four groups), respectively. In addition, a comprehensive analysis of the elongation of a specimen with Parameter 3 in Figure 21 shows that the values for elongation in the three directions of ‘H’, ‘T’ and ‘V’ were 56.07%, 53.07%, and 55.73%, respectively, all of which are relatively high. Although the elongation of Parameter 1 is slightly higher, the specimen exhibits some anisotropy in the three directions of Parameter 1. The elongation in the ‘H’ direction is 61.07%, which is 5% higher than the other two directions. Therefore, from a comprehensive analysis perspective, the mechanical properties of Parameter 3 specimen are superior.

#### 3.3.2. Optimal Combination of Scan Spacing

Parameters 4, 5, 6, and 7 in Table 3 were adopted to investigate the influence of different scanning spacings on the formability of SLM 316L SS. For these four sets of parameters, the laser power and scanning speed were set at 300 W and 1000 mm/s. The scanning spacings for Parameters 4, 5, 6, and 7 were 0.10 mm, 0.12 mm, 0.14 mm, and 0.08 mm, respectively. The optimal scanning spacing was determined by analyzing the tensile properties of specimens with different scanning spacings. The true stress-strain curves for Face 3 of Parameters 4, 5, and 7, and Face 2 of Parameter 6 can be obtained from Figure 8, Figure 9, Figure 10 and Figure 11. Figure 22 shows the elongation of the corresponding specimens.

From the perspective of the proximity of stress-strain curves, the stress-strain curves of Parameters 5 and 7 in three directions are closer. Therefore, the specimens printed using Parameters 5 and 7 tend to be more isotropic. As shown in Figure 22, although the elongation in the H and V directions of Parameter 5 is significantly different from that in the T direction, it is evidently superior to that of the printed specimen with Parameter 7. In addition, according to the mechanical properties’ indexes, the average yield strength, tensile strength, and elastic modulus of specimens in different directions of Parameter 5 reached 433.95 MPa (the second in the four groups, with a difference of 10 MPa from the first), 819.03 MPa (the largest in the four groups), and 128.30 MPa (the largest in the four groups), respectively. Therefore, from a comprehensive analysis perspective, the mechanical properties of Parameter 5 specimen are superior.

#### 3.3.3. Optimal Combination of Scanning Speed

Parameters 7, 8, 9, and 10 in Table 3 were adopted to investigate the impact of different scanning speeds on the formability of SLM 316L SS. Among the four sets of parameters, the laser power and scanning spacing were set at 300 W and 0.08 mm. The scanning speeds of Parameters 7, 8, 9, and 10 were 1000 mm/s, 950 mm/s, 900 mm/s, and 850 mm/s, respectively. The optimal scanning speeds were determined by analyzing the tensile properties of specimens at different scanning speeds. The true stress-strain curve for Face 3 with Parameters 7, 8, 9, and 10 can be obtained from Figure 11, Figure 12, Figure 13 and Figure 14. Figure 23 shows the elongation of the corresponding specimens.

From the perspective of the proximity of stress-strain curves, the stress-strain curves of Parameters 7 and 10 in three directions are closer. Therefore, the specimens printed using Parameters 7 and 10 tend to be more isotropic. It can be observed from Figure 23 that the tensile properties of the specimen printed with Parameter 10 exhibit less variation in three directions and are significantly superior to the elongation of the specimen printed with Parameter 7. In terms of mechanical property indices, the average yield strength, tensile strength, and elastic modulus of specimens in different directions of Parameter 10 reached 453.38 MPa (the largest in the four groups), 852.54 MPa (the largest in the four groups), and 127.74 MPa (the smallest in the four groups, but differs little from the elastic modulus of the other three parameters, and differs by only 9 GPa from the maximum), respectively. Therefore, from a comprehensive analysis perspective, the mechanical properties of Parameter 10 specimen are superior.

Based on the analysis above, the optimal process parameter combination for the SLM 316L SS specimen in the printing parameters used in this paper are Process Parameters 3, 5, and 10.

In this study, no clear linear relation between the strength of mechanical properties and printing parameters are found. This lack of relation may be attributed to the small gradient set between each group of process parameters, which makes it difficult to discern the influence of each printing parameter on the mechanical properties of the specimen.

## 4. Conclusions

(1)The anisotropy of SLM 316L SS is primarily evident in Faces 1 and 2 (parallel to the building direction), while Face 3 (perpendicular to the building direction) shows good isotropy.(2)The mechanical properties of specimens with different directions in the same face were analyzed. Overall, it can be observed that the mechanical properties in H and T directions are similar, while the mechanical properties in V direction are inferior. The order of mechanical properties from good to bad in Faces 1 and 2 is T, H, and V; the mechanical properties in the H, T, and V directions in Face 3 are similar.(3)The mechanical properties of the specimen with the same direction in different faces were analyzed. Overall, it can be observed that the strength of Faces 1 and 2 are similar, while the strength of Face 3 is higher. Among them, the performance of the H direction for the three faces is relatively close; in the T direction, Face 1 is slightly stronger than Faces 2 and 3, while Face 2 is relatively close to Face 3. The mechanical properties of Face 3 in the V direction are the best, and the V directions of Faces 1 and 2 are relatively close.(4)The superior mechanical properties of Face 3 are attributed to its smaller grain size and higher dislocation density. The anisotropy of the mechanical properties of Face 1 and Face 2 are attributed to the presence of texture and columnar crystals in the sample.(5)The optimal process parameter combination for the SLM 316L SS specimen, based on the printing parameters used in this paper, is Parameters 3, 5, and 10.

This paper provides a comprehensive description of the anisotropy of mechanical properties of SLM 316L SS, offering valuable guidance for engineering design and practical manufacturing process optimization. Additionally, it offers essential references for researchers seeking to select the most suitable processing parameters. Future studies should prioritize the investigation of anisotropy in hardness and fatigue properties to further enhance the understanding and utilization of SLM 316L SS.

## Figures and Tables

**Figure 1 materials-17-02017-f001:**
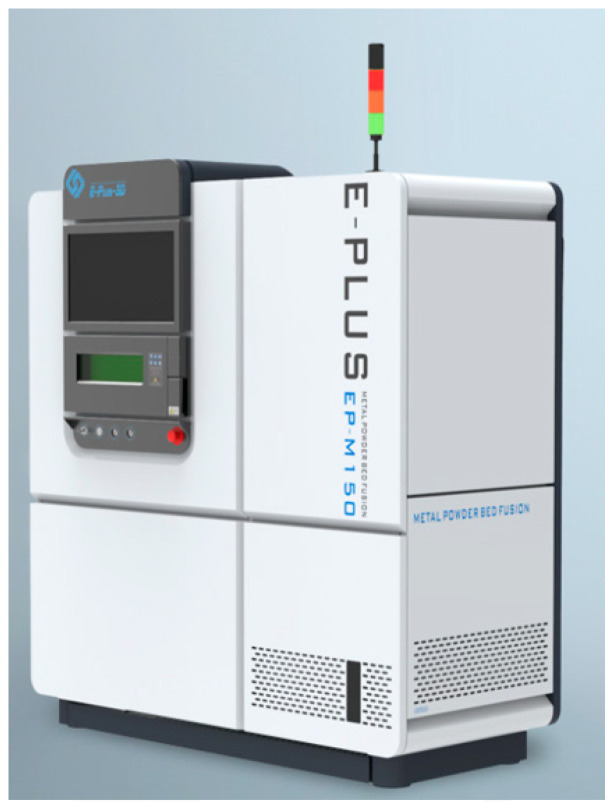
EP-M150 metal additive manufacturing equipment.

**Figure 2 materials-17-02017-f002:**
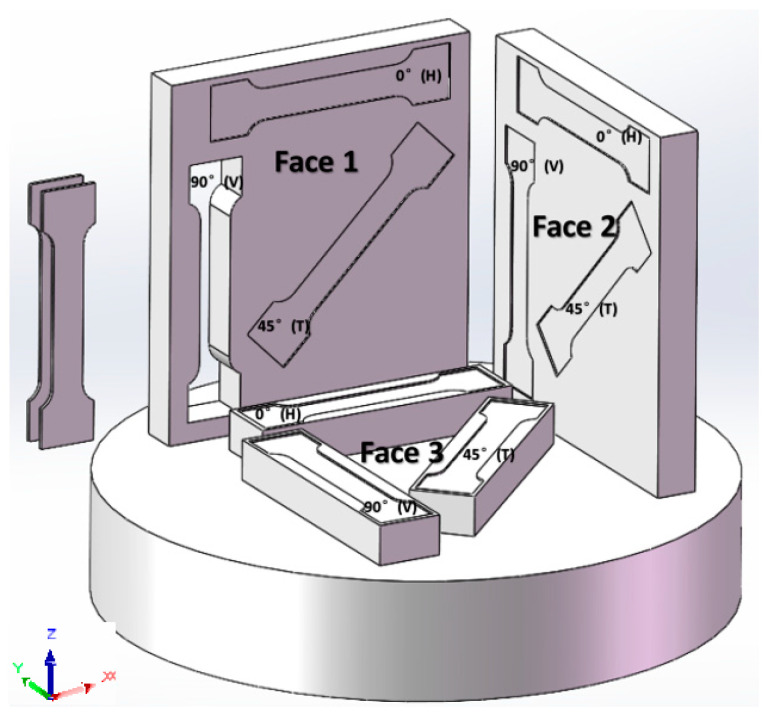
The printed sample and pick-up position.

**Figure 3 materials-17-02017-f003:**
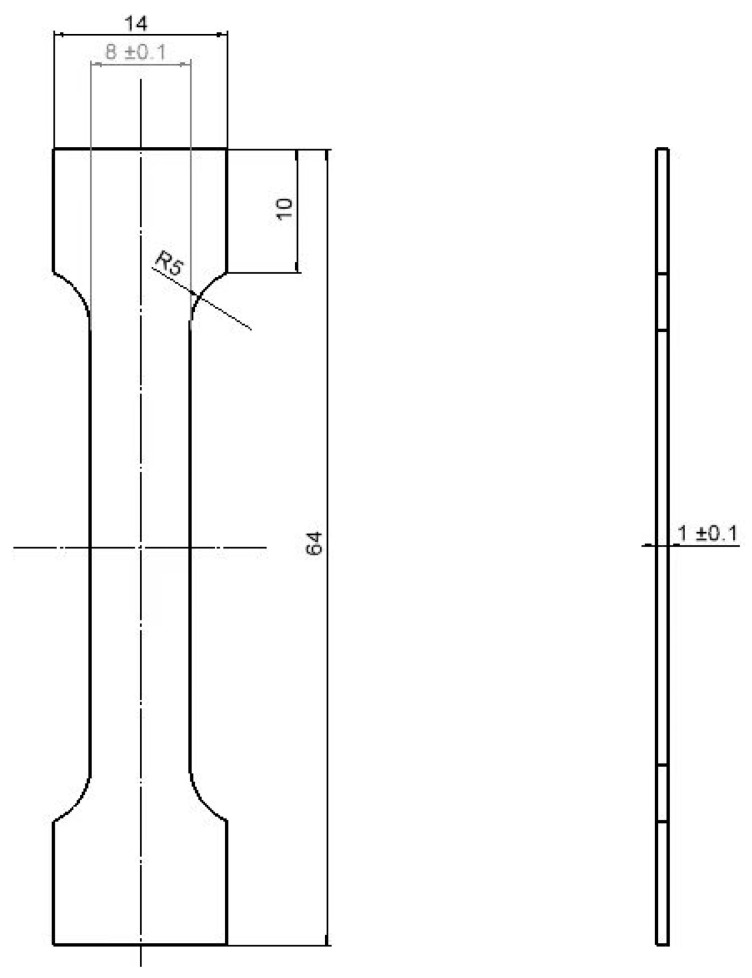
The specific size of uniaxial tensile specimen, Unit: mm.

**Figure 4 materials-17-02017-f004:**
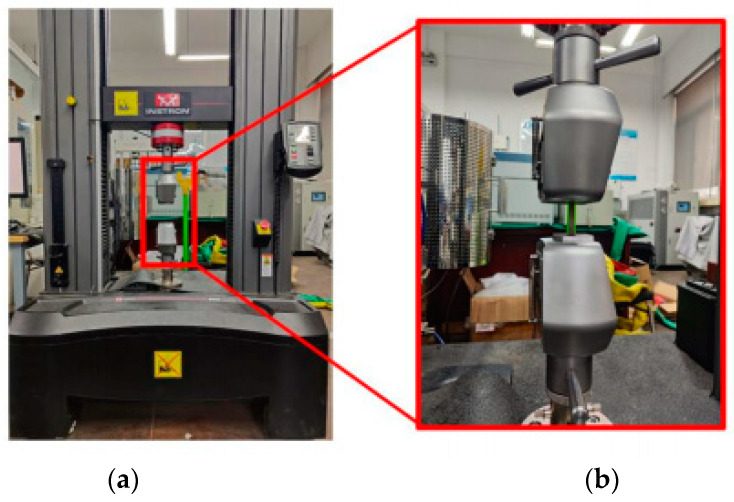
Instron universal testing machine. (**a**) Instron universal testing machine; (**b**) tester collet.

**Figure 5 materials-17-02017-f005:**
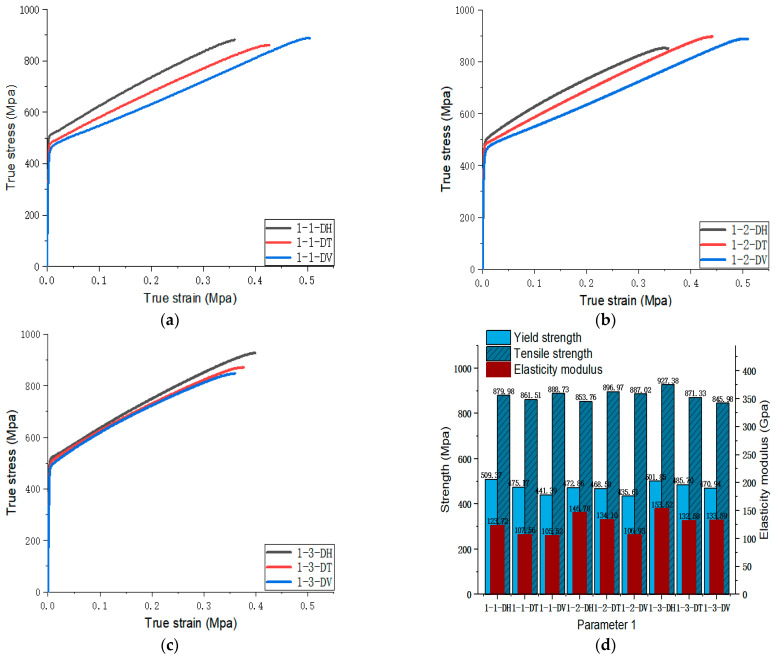
Parameter 1: (**a**) true stress-strain curves of three specimens on Face 1; (**b**) true stress-strain curves of three specimens on Face 2; (**c**) true stress-strain curves of three specimens on Face 3; (**d**) the yield strength, tensile strength, and elastic modulus of the specimen with parameter 1.

**Figure 6 materials-17-02017-f006:**
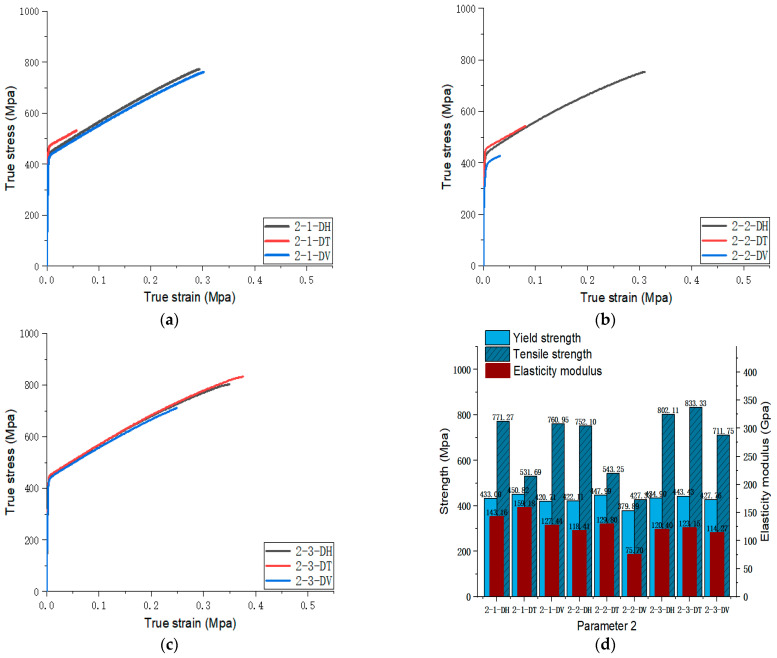
Parameter 2: (**a**) true stress-strain curves of three specimens on Face 1; (**b**) true stress-strain curves of three specimens on Face 2; (**c**) true stress-strain curves of three specimens on Face 3; (**d**) the yield strength, tensile strength, and elastic modulus of the specimen with parameter 2.

**Figure 7 materials-17-02017-f007:**
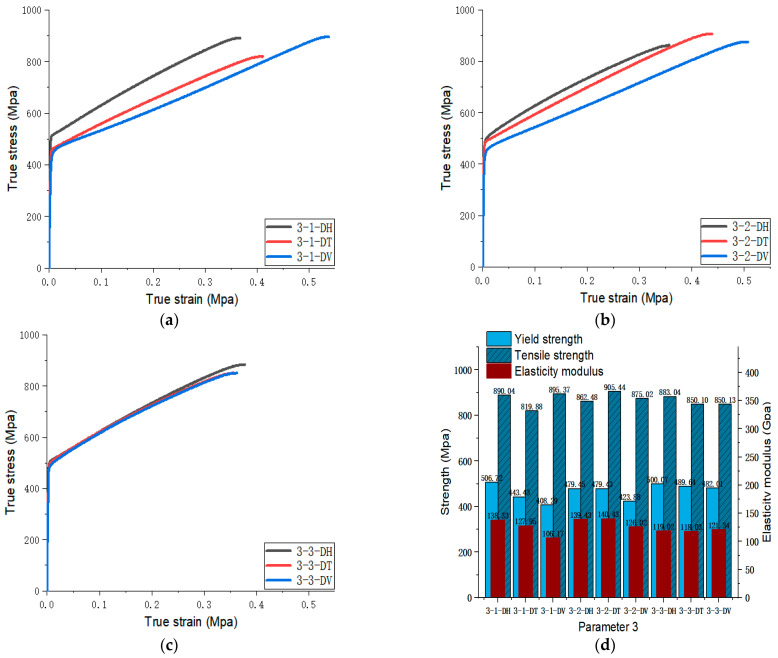
Parameter 3: (**a**) true stress-strain curves of three specimens on Face 1; (**b**) true stress-strain curves of three specimens on Face 2; (**c**) true stress-strain curves of three specimens on Face 3; (**d**) the yield strength, tensile strength, and elastic modulus of the specimen with parameter 3.

**Figure 8 materials-17-02017-f008:**
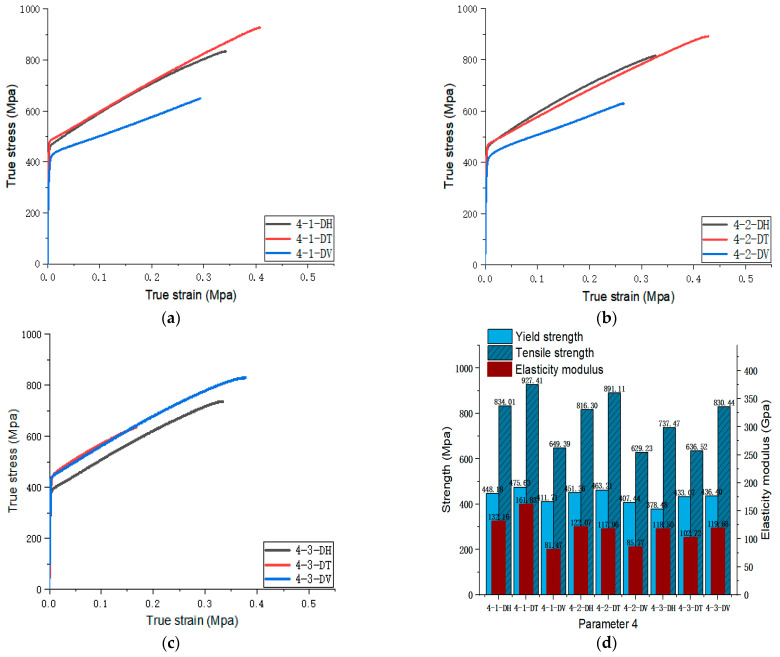
Parameter 4: (**a**) true stress-strain curves of three specimens on Face 1; (**b**) true stress-strain curves of three specimens on Face 2; (**c**) true stress-strain curves of three specimens on Face 3; (**d**) the yield strength, tensile strength, and elastic modulus of the specimen with parameter 4.

**Figure 9 materials-17-02017-f009:**
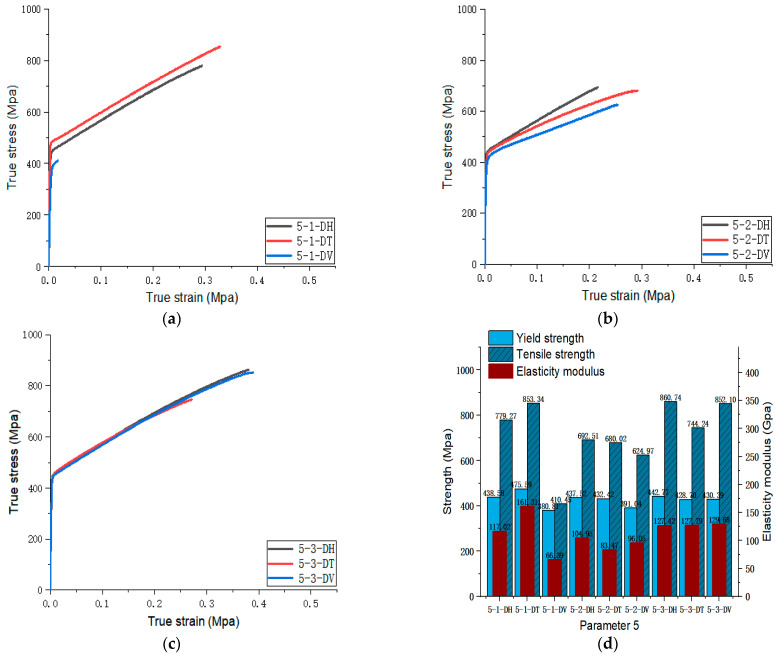
Parameter 5: (**a**) true stress-strain curves of three specimens on Face 1; (**b**) true stress-strain curves of three specimens on Face 2; (**c**) true stress-strain curves of three specimens on Face 3; (**d**) the yield strength, tensile strength, and elastic modulus of the specimen with parameter 5.

**Figure 10 materials-17-02017-f010:**
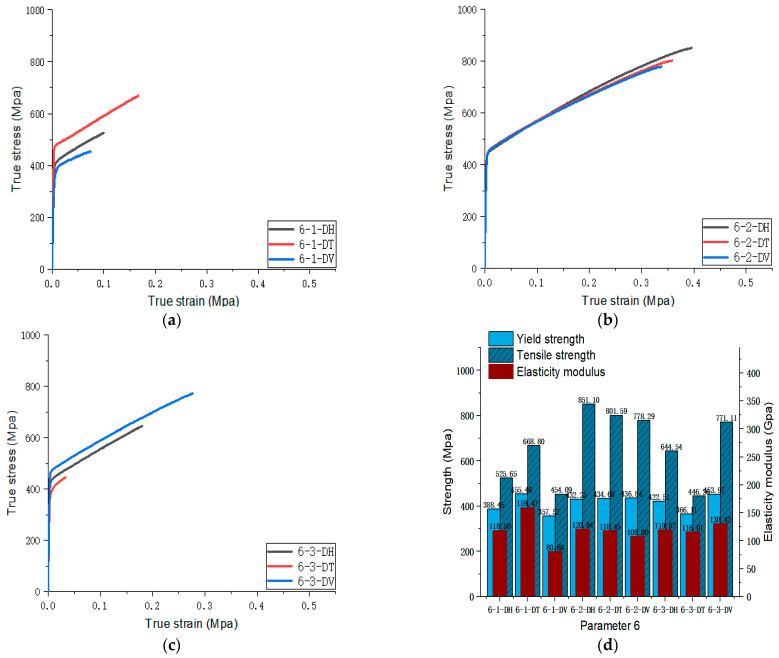
Parameter 6: (**a**) true stress-strain curves of three specimens on Face 1; (**b**) true stress-strain curves of three specimens on Face 2; (**c**) true stress-strain curves of three specimens on Face 3; (**d**) the yield strength, tensile strength, and elastic modulus of the specimen with parameter 6.

**Figure 11 materials-17-02017-f011:**
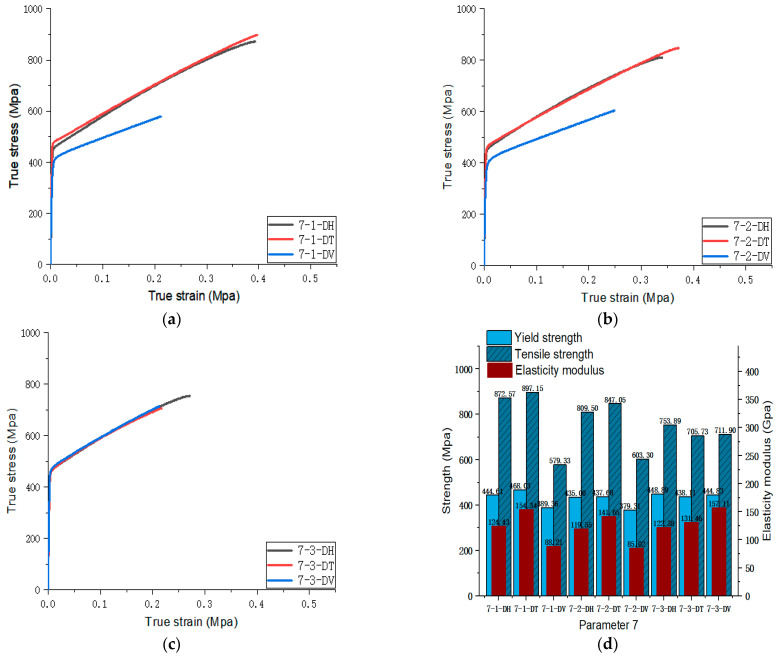
Parameter 7: (**a**) true stress-strain curves of three specimens on Face 1; (**b**) true stress-strain curves of three specimens on Face 2; (**c**) true stress-strain curves of three specimens on Face 3; (**d**) the yield strength, tensile strength, and elastic modulus of the specimen with parameter 7.

**Figure 12 materials-17-02017-f012:**
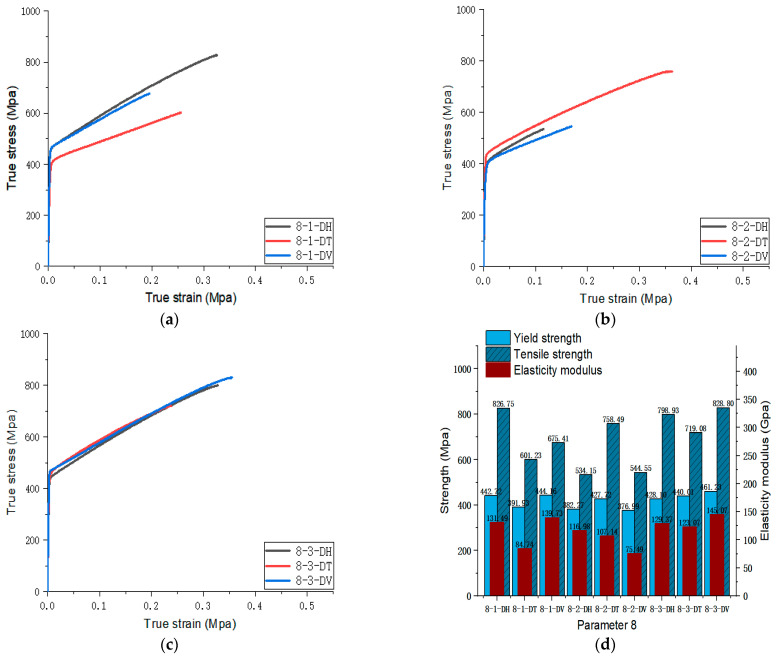
Parameter 8: (**a**) true stress-strain curves of three specimens on Face 1; (**b**) true stress-strain curves of three specimens on Face 2; (**c**) true stress-strain curves of three specimens on Face 3; (**d**) the yield strength, tensile strength, and elastic modulus of the specimen with parameter 8.

**Figure 13 materials-17-02017-f013:**
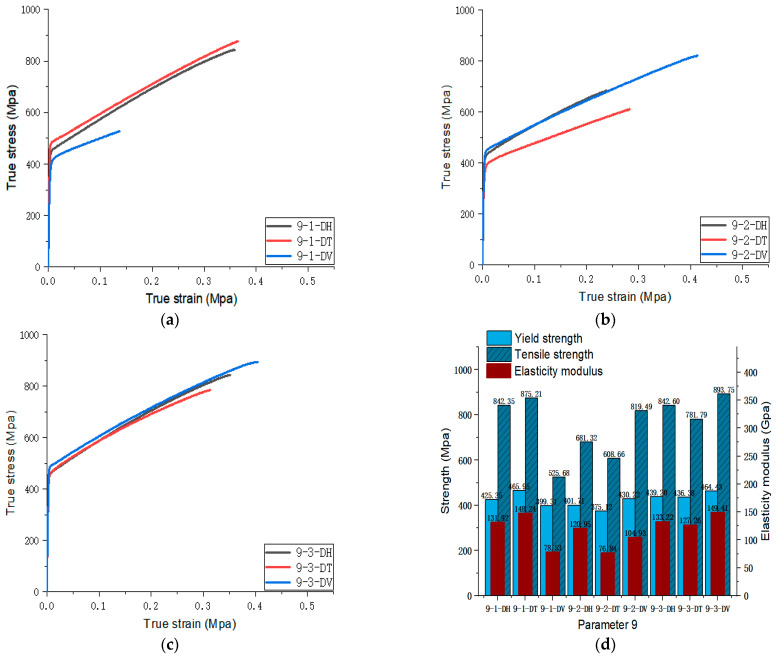
Parameter 9: (**a**) true stress-strain curves of three specimens on Face 1; (**b**) true stress-strain curves of three specimens on Face 2; (**c**) true stress-strain curves of three specimens on Face 3; (**d**) the yield strength, tensile strength, and elastic modulus of the specimen with parameter 9.

**Figure 14 materials-17-02017-f014:**
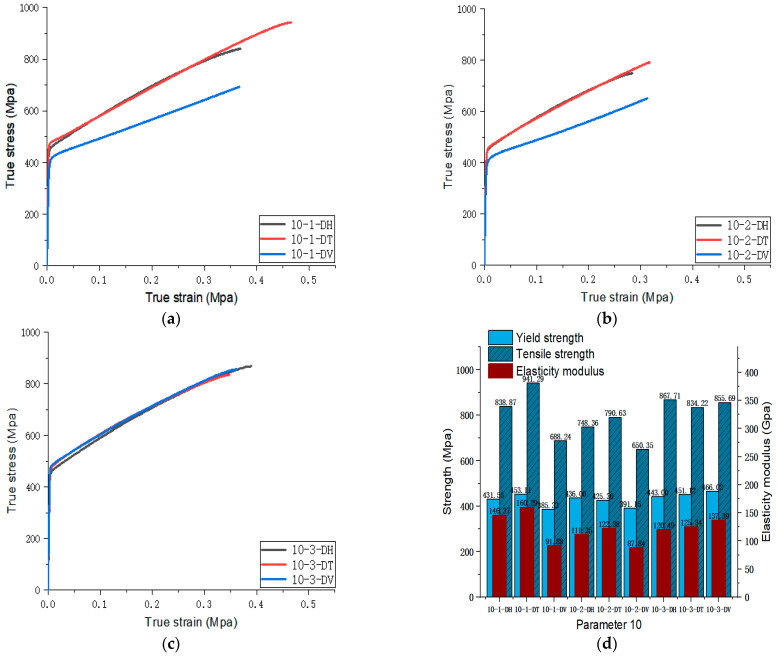
Parameter 10: (**a**) true stress-strain curves of three specimens on Face 1; (**b**) true stress-strain curves of three specimens on Face 2; (**c**) true stress-strain curves of three specimens on Face 3; (**d**) the yield strength, tensile strength, and elastic modulus of the specimen with parameter 10.

**Figure 15 materials-17-02017-f015:**
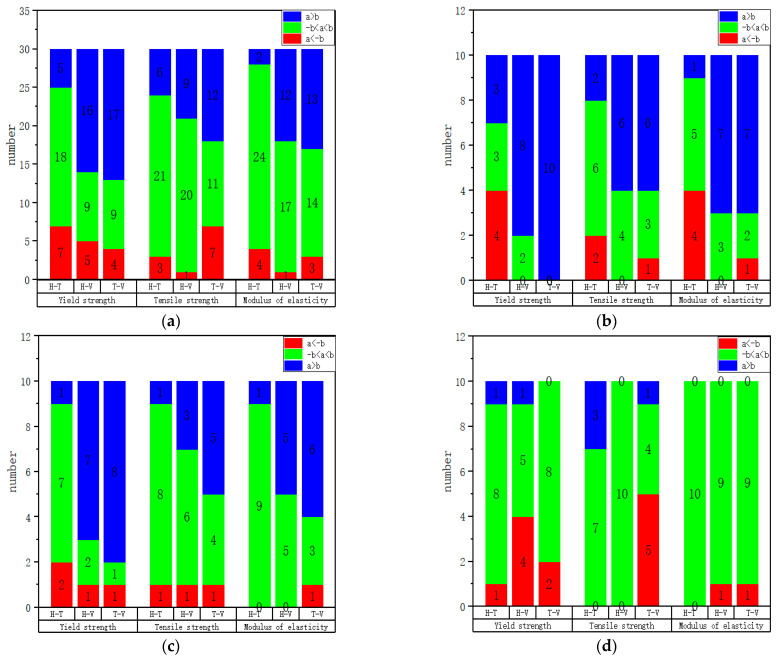
Statistical results of mechanical properties comparison of specimens with different directions on the same face: (**a**) complete specimens; (**b**) Face 1; (**c**) Face 2; (**d**) Face 3.

**Figure 16 materials-17-02017-f016:**
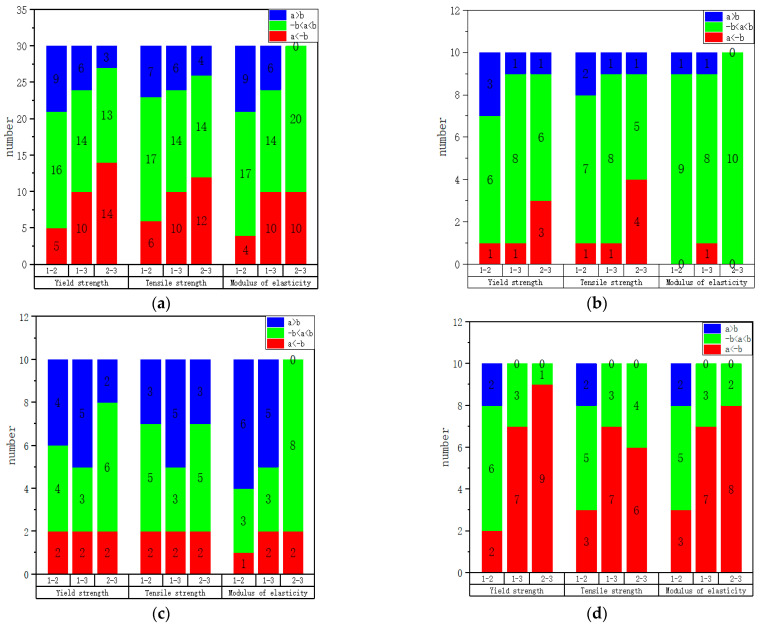
Statistical results of mechanical properties comparison of specimens with different faces in the same direction: (**a**) complete specimens; (**b**) H direction; (**c**) T direction; (**d**) V direction.

**Figure 17 materials-17-02017-f017:**
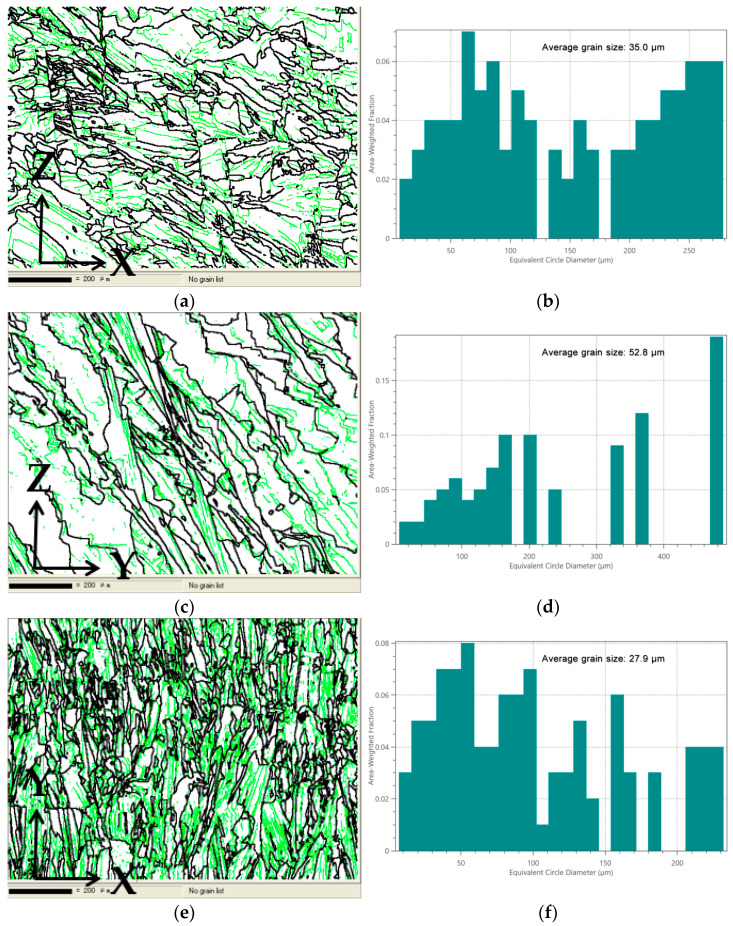
Microscopic analysis: (**a**) grain boundary figure of Face 1; (**b**) grain size distribution of Face 1; (**c**) grain boundary figure of Face 2; (**d**) grain size distribution of Face 2; (**e**) grain boundary figure of Face 3; (**f**) grain size distribution of Face 3; (**g**) kernel average misorientation figure of Face 1; (**h**) kernel average misorientation figure of Face 2; (**i**) kernel average misorientation figure of Face 3.

**Figure 18 materials-17-02017-f018:**
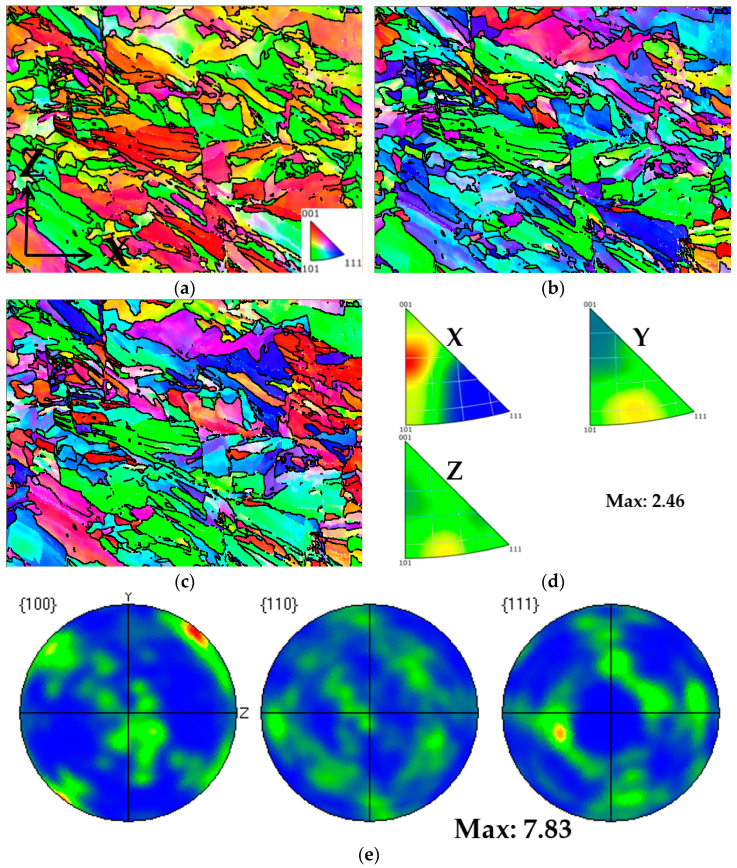
Texture analysis of Face 1: (**a**) IPF coloring map parallel to X; (**b**) IPF coloring map parallel to Y; (**c**) IPF coloring map parallel to Z; (**d**) IPF maps; (**e**) pole figures.

**Figure 19 materials-17-02017-f019:**
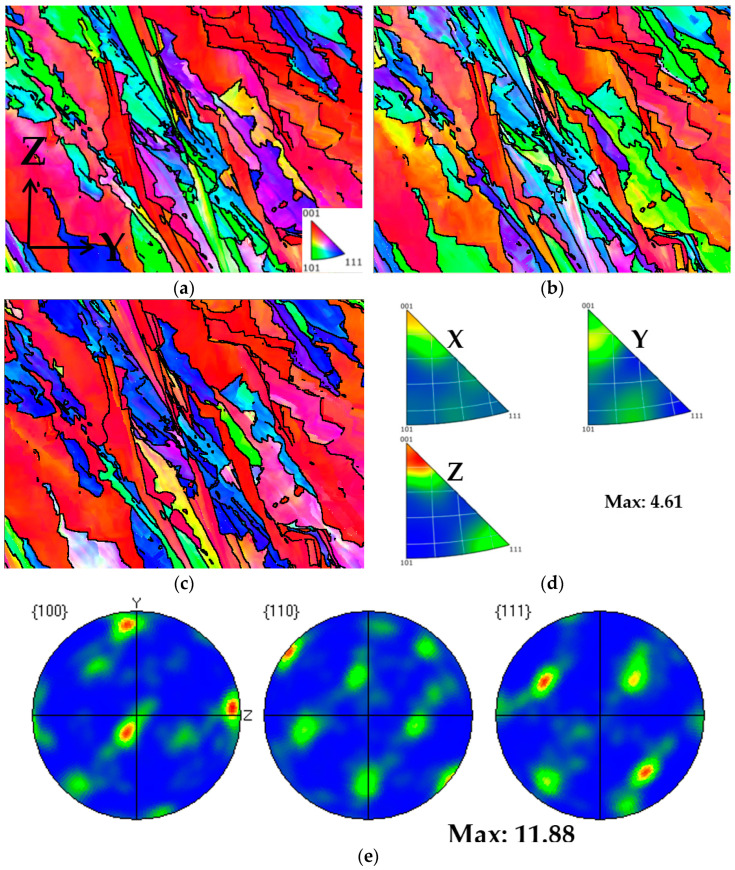
Texture analysis of Face 2: (**a**) IPF coloring map parallel to X; (**b**) IPF coloring map parallel to Y; (**c**) IPF coloring map parallel to Z; (**d**) IPF maps; (**e**) pole figures.

**Figure 20 materials-17-02017-f020:**
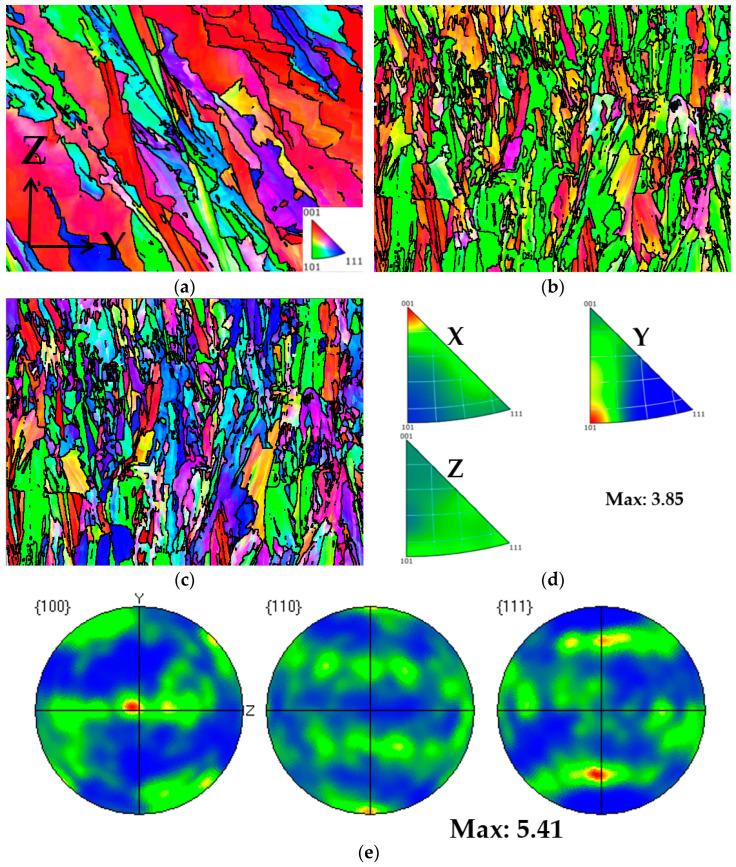
Texture analysis of Face 3: (**a**) IPF coloring map parallel to X; (**b**) IPF coloring map parallel to Y; (**c**) IPF coloring map parallel to Z; (**d**) IPF maps; (**e**) pole figures.

**Figure 21 materials-17-02017-f021:**
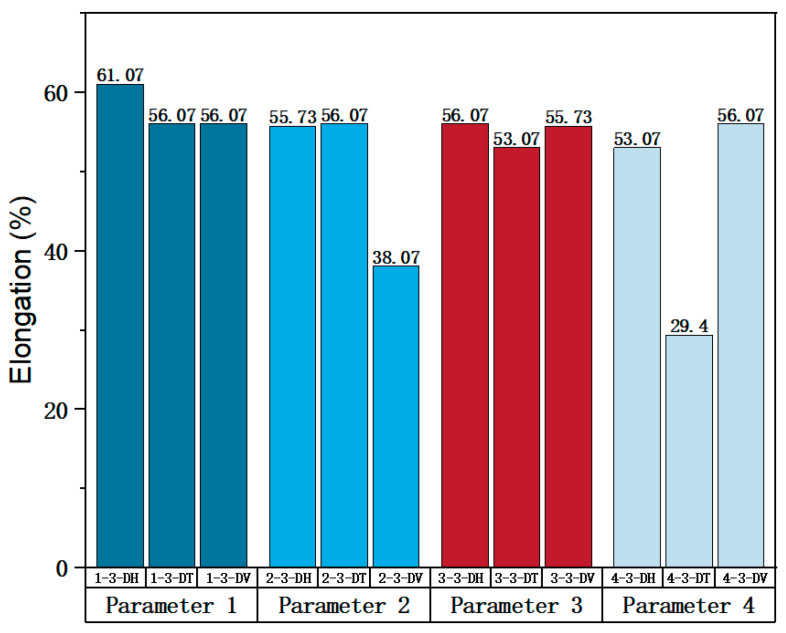
Elongation of the corresponding specimens.

**Figure 22 materials-17-02017-f022:**
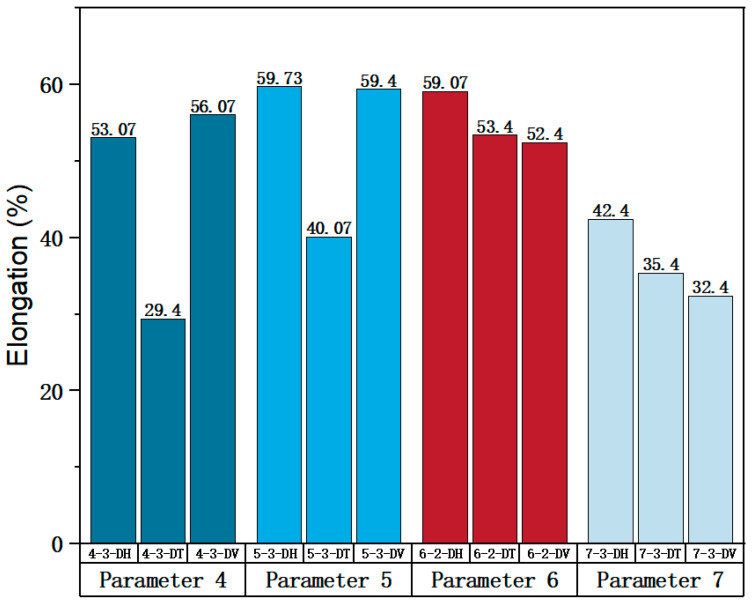
Elongation of the corresponding specimens.

**Figure 23 materials-17-02017-f023:**
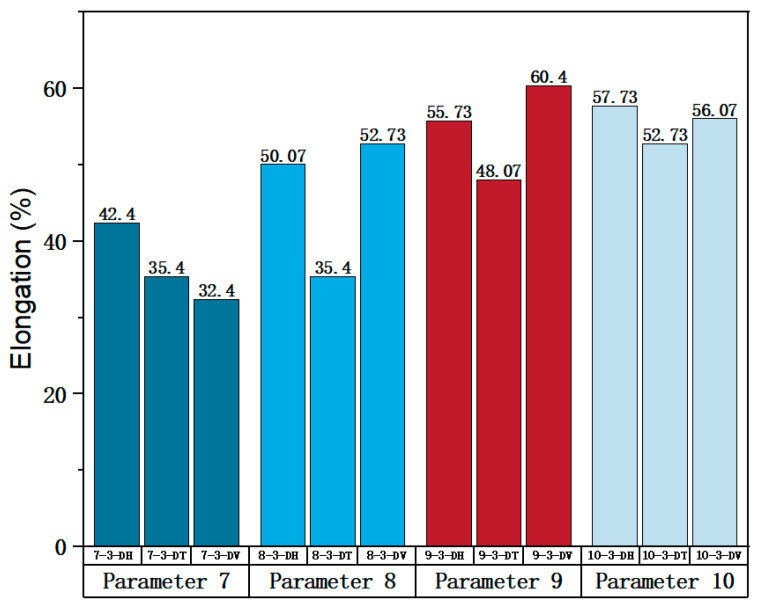
Elongation of the corresponding specimens.

**Table 1 materials-17-02017-t001:** Chemical composition of 316L SS powder.

Element	C	Mn	Si	Cr	Ni	Mo	S	P	Fe
Wt. (%)	0.004	0.99	0.63	16.79	11.78	2.43	0.001	0.022	balance

**Table 2 materials-17-02017-t002:** Physical properties of 316L SS powder.

Material	Particle Size Distribution (um)				Bulk Density	Fluidity	Oxygen Content	Sphericity
	D10	D50	D90	D97	(g/cm^3^)	(s/50g)	(ppm)	
316L SS	18.04	33.72	59.98	75.46	4.37	13.34	330	0.881

**Table 3 materials-17-02017-t003:** SLM process parameter setting.

Number	Laser Power (W)	Scanning Spacing (mm)	Scanning Speed (mm/s)
Parameter 1	200	0.10	1000
Parameter 2	230	0.10	1000
Parameter 3	270	0.10	1000
Parameter 4	300	0.10	1000
Parameter 5	300	0.12	1000
Parameter 6	300	0.14	1000
Parameter 7	300	0.08	1000
Parameter 8	300	0.08	950
Parameter 9	300	0.08	900
Parameter 10	300	0.08	850

## Data Availability

Data are available from the corresponding author upon request.

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
