# Peer review of "An Investigation of the Anisotropic Mechanical Properties of Additive-Manufactured 316L SS with SLM"

_materials, 2024, doi:10.3390/ma17092017_

Round 1
Reviewer 1 Report
Comments and Suggestions for Authors

There are a few instances where sentence structure and grammar could be refined for better readability. Please consider reviewing the paper for subject-verb agreement, punctuation, and overall sentence structure to ensure a smoother flow.
Reviewer 2 Report
Comments and Suggestions for Authors
The manuscript entitled “Investigation of anisotropic mechanical properties of addictive manufactured 316L SS with SLM” was reviewed. The work carried out in the manuscript is interesting. However, in my opinion, several aspects should be modified or detailed more in-depth prior to publication, thus major modifications are advised. Here is a list of main comments:
Detailed comments:
-It is highly recommended to provide a graphical abstract, as it will increase the visibility of the work and make the manuscript more appealing.
- Please, demonstrate the novelty and importance of this paper in the abstract and at the end of the introduction.
- The title effectively summarizes the content of the paper, which is about investigating anisotropic mechanical properties of additively manufactured 316L stainless steel using Selective Laser Melting (SLM).
- The abstract is clear and concise, providing a good overview of the paper's purpose and findings. It introduces the concept of anisotropic mechanical properties in SLM-formed specimens and mentions the focus on different faces and orientations. However, the abstract could be improved by briefly mentioning the key findings.
-The introduction provides essential background information on 316L SS and the significance of SLM in its manufacturing. It cites relevant studies on mechanical anisotropy in AM specimens and sets the stage for the paper's objectives.
-Research Gap: It correctly identifies the gap in existing research related to mechanical properties in different directions and faces for SLM-formed specimens.
-While the significance of this study is evident, it might be beneficial to emphasize how understanding anisotropic behavior in SLM-formed 316L SS specimens can lead to more efficient use of this manufacturing method in various applications.
-The paper provides a comprehensive overview of previous studies related to mechanical anisotropy in AM specimens. However, it would be more helpful to briefly highlight the key findings of these studies.
-How was the laser energy density adjusted or controlled during the experiments?
-Were there any specific reasons for choosing the three tested directions (0°, 45°, 90°) and faces (XY, XZ, YZ)? Did these choices relate to real-world applications or industry needs?
-Could you clarify the relevance of energy density in the context of SLM and its impact on mechanical properties?
-It might be beneficial to briefly mention the relevance of particle size distribution, fluidity, and bulk density to the study. These properties are important factors, and it would help the reader understand their role.
-The paper discusses anisotropic properties of SLM 316L SS, with Faces 1 and 2 showing more anisotropy than Face 3. Could you provide a deeper understanding of why this anisotropy occurs? Are there specific microstructural or process-related explanations for this behavior?
-In section 3.2, you identified optimal process parameters as Parameters 3, 5, and 10. Can you discuss potential trade-offs or limitations associated with these parameter choices, such as production time, cost, or other factors?
-The study examined the effect of laser power, scanning speed, and scanning spacing on mechanical properties. Did you observe any interactions or synergistic effects between these parameters? For example, does changing one parameter influence the optimal setting of another?
-The research focused on mechanical properties like yield strength, tensile strength, and elastic modulus. Were there any other properties or characteristics of SLM 316L SS that could be affected by the selected process parameters, and should they be considered in future studies?
-Your paper emphasizes the importance of laser energy density in SLM. Could you elaborate on the role of laser energy density in the context of powder melting and its implications for part quality and defects?
-Were there any challenges or limitations you encountered during the experiment, and how did you address them? For instance, did you face any issues related to the quality and consistency of the 3D printing process, or data collection and analysis?
-Finally, in the conclusion section you need to highlight that, with the optimal process parameters identified in your study, what are the next steps or recommendations for using SLM 316L SS in practical applications, and what areas of further research do you believe are crucial for enhancing the understanding and use of this material?
Reviewer 3 Report
Comments and Suggestions for Authors
The authors have embarked on a promising exploration of the anisotropic mechanical properties of additive manufactured 316L SS with SLM. While this research holds immense potential, it appears to be in its early stages, and substantial work is required before it reaches a publishable standard. Here are some constructive comments intended to inspire and enhance the work conducted by the authors thus far. These suggestions aim to guide the authors in refining their research, thereby contributing to the advancement of knowledge in this field:
The manuscript’s most significant limitation is the absence of microstructural characterization, which is crucial for understanding the impact of the various parameters evaluated. Without this evidence, comprehending the differences among the directions or parameters used during the SLM becomes a Herculean task. A comprehensive microstructural study employing Optical microscopy, SEM, EDS (chemical analysis), and EBSD was anticipated. Such a study would illuminate the diverse results obtained during the tensile tests. For example, without microstructural characterization, it’s impossible to discern and understand the influence of scanning speed, laser power, energy density, and other parameters on the mechanical properties. Therefore, it is imperative to include microstructural characterization. This will enable future readers to draw meaningful correlations between mechanical properties and process parameters, and concurrently observe the microstructural changes. This addition is a must and significantly enhance the manuscript’s value and impact.
The methodology section could be enhanced with additional details to provide a more comprehensive understanding. For instance, the strain rate used during the tensile tests is not specified. It would be beneficial to know if the ASTM standard E8 was adhered to during these tests. Furthermore, the number of samples tested in each group as presented in Table 3 is unclear and the standard deviation of those results. The paragraph “Go along the thickness direction to cut out the required specimens, two in each direction, and finally six single tensile specimens were obtained for each face. In the drawing of stress-strain diagrams in Chapter 3, we used the two specimens with more complete stress-strain diagrams for the study” it not clears at all. The authors are referring to a Chapter 3, that there is not available in this manuscript, the number of tests per condition it is not clear and creates confusion.
The manuscript’s writing style is challenging to follow, which hampers readability. For instance, the sentence “The numbering rule for unidirectional tensile specimens is the ‘process parameters -taking the plane - taking the direction’. Take ‘1-1-DH’ for example, in which the first number is the serial number of process parameter setting in Table 3, and ‘1’ indicates the parameter 1; the second number indicates the plane of specimen taking, and the XY Face perpendicular to the stacking direction is called ‘Face 3’, and the YZ and XZ Faces parallel to the stacking direction are referred to as ‘Face 1’ and ‘Face 2’, so the second digit ‘1’ represents ‘Face 1’;” requires careful revision throughout the manuscript to facilitate easier interpretation by readers. This includes modifying and revising Figures 5 to 14, which are currently too blurry. The quality of these images should be improved for clarity. Readers should be able to easily understand terms like ‘1-DH’, ‘1-DT’, and ‘1-DV’. Similarly, the authors should include in the legend of each Figure the meaning of the plots a, b, c, and d. Including the meaning of the abbreviations inside the plots. Similarly, each figure’s legend should include an explanation of plot meanings a, b, c, and d, including any abbreviations used within the plots. The x and y-axis scales should be kept consistent across all plots; some currently go up to 1000 MPa, others up to 800 MPa, and some exceed 1000 MPa. Please revise accordingly.
The introduction section should be revised carefully since the reader it is expected to be guide by the authors by showing the state-of-the-art literature and some discussion of these works. In other words, the works should not be quoted with the main mechanical properties obtained in each direction but also the authors should discuss briefly why these results were obtained or mentioned what is missing in those works. In the sentence: “Some other scholars have studied the mechanical properties of AM specimens with different faces and orientations of the specimens. In their study of SLM Ti6Al4V specimens, Beibei He et al. found that the XZ plane of the Ti6Al4V…” the authors should revise the word “scholars” since does not seem to be suitable in this context, and in addition, the authors mentioned in this sentence “their”, without refereeing what the authors are referring to. The reader assumes is “Beibei He et al.” but the phrase should be revised and include the proper quotation of that work. Similar comments about the lack of reference applies to “On the anisotropy of thick-walled wire arc AM stainless steel parts, L. Palmeira Belotti et al. found that under uniaxial tensile loading”, and “L. Hitzler et al. found the general rule of higher”, and “R. Nandhakumar et al. summarized the findings”, and “Wakshum M. Tucho et al. found that 316L”. On the other hand, is it a typo in “Barıs,S, ener [25]”? In the sentence “the yield strength of SLM-formed 316L is higher than that of the conventional method.” What is the conventional method? Please specify.
The authors commonly use the word “Face” with capital letters; however, it is no clear what this mean.
The equations from 1-4 are well known. Nonetheless, the authors should include a suitable reference of these.
The authors’ discussion on anisotropy and the mechanical properties associated with the building direction in the additive manufacturing process is insightful. However, given that this process involves a specific axis of layer building and that the tensile tests require an understanding of the tensile direction, it is crucial to clarify these aspects. Therefore, I strongly urge the authors to revise the schemes in Figures 3 and 4. Providing a clearer understanding of the direction, including the three different angles/orientations, would significantly enhance the manuscript’s comprehensibility. This revision would not only make your work more accessible to readers but also underscore the importance and relevance of your research in this field.
Comments on the Quality of English LanguageThe coherence among sentences and the overall articulation in the text require significant enhancement. It’s crucial to ensure that each sentence logically connects to the next, creating a smooth flow of ideas. This will not only improve readability but also strengthen the impact of the message being conveyed. A well-articulated piece of writing engages the reader, making the content more memorable and persuasive. Therefore, investing time in refining sentence structure, improving transitions, and clarifying ideas can greatly elevate the quality of the text.
Reviewer 4 Report
Comments and Suggestions for Authors
Two basic problems make the manuscript need to be revised. The first is because of the English language, which is generally good but with many small mistakes (given in the pdf file). The second is the poor visibility of some of the images (details in the pdf file). The text has only a few inconsistencies, but they can be quickly corrected. There are also a few issues with inconsistent formatting.
Although numerous corrections are needed, they are not fundamental, and a minor revision is required before publishing.

The English language is excellent, but there are still many mistakes, such as an article omitted or added unnecessarily, singular/plural, missing commas in a series, etc.
Round 2
Reviewer 1 Report
Comments and Suggestions for Authors
No comments.
Comments on the Quality of English LanguageNo comments
Author Response
Dear Reviewer,
Thank you for your review of our article and valuable comments. We have modified the full text of English. The second revision of the article we highlighted with green.
If there are any other problems or questions about our manuscript, please do not hesitate to let us know.
Yours sincerely,
Peng Jiang
January 9, 2024
Reviewer 2 Report
Comments and Suggestions for Authors
The requested modifications have been done correctly, congratulations to the authors.
Author Response
Dear Reviewer,
Thank you very much for your review of our articles and recognition of our work.
Here to express my most sincere thanks to you.
Yours sincerely,
Peng Jiang
January 9, 2024